# Immune Deficiency/Dysregulation-Associated EBV-Positive Classic Hodgkin Lymphoma

**DOI:** 10.3390/cancers17091433

**Published:** 2025-04-25

**Authors:** Mohamed Nazem Alibrahim, Annunziata Gloghini, Antonino Carbone

**Affiliations:** 1Faculty of Medicine, Zagazig University, Zagazig 44519, Egypt; 2Department of Avanced Pathology, Fondazione Istituto di Ricovero e Cura a Carattere Scientifico IRCCS, Istituto Nazionale dei Tumori Milano, 20133 Milano, Italy; annunziata.gloghini@istitutotumori.mi.it; 3Centro di Riferimento Oncologico, Istituto di Ricovero e Cura a Carattere Scientifico, National Cancer Institute, 33081 Aviano, Italy

**Keywords:** classic Hodgkin lymphoma, EBV+ cHL, WHO classification, immune deficiency, immune dysregulation, management, diagnosis

## Abstract

Classic Hodgkin lymphoma (cHL) associated with immune deficiency or dysregulation, particularly involving Epstein–Barr virus (EBV), represents a distinct and challenging clinical entity. EBV plays a crucial role by promoting cancer cell survival, altering immune responses, and fostering immune escape mechanisms. This subtype of lymphoma frequently arises in immunocompromised individuals, such as HIV-positive patients, transplant recipients, and those with primary immunodeficiencies, and is often resistant to standard treatments. Current classifications recognize the importance of EBV status and immune dysfunction in diagnosis and therapy planning. New strategies, including immune checkpoint inhibitors, EBV-targeted T-cell therapies, and novel molecular approaches, show promise in improving outcomes. Understanding the complex interplay between viral infections, immune status, and genetics is essential for developing targeted treatments and personalized patient care.

## 1. Introduction

Hodgkin lymphoma is categorized into two primary forms, each with unique pathological and clinical characteristics: classic Hodgkin lymphoma (cHL) and nodular lymphocyte-predominant Hodgkin lymphoma (NLPHL). cHL is further divided into four subtypes, nodular sclerosis (NSHL), lymphocyte-rich (LRHL), mixed cellularity (MCHL), and lymphocyte-depleted (LDHL) [1]. cHL is defined by the presence of Reed–Sternberg (RS) cells, large, often multinucleated malignant cells with characteristic “owl-eye” nuclei, originating from germinal center (GC) B cells with immunoglobulin gene rearrangements. These cells are typically positive for CD30 and CD15, but negative for B cell markers CD20 and CD45, and are often accompanied by a rich inflammatory background shaped by cytokines secreted by RS cells. Variants of RS cells include lacunar cells, mummified cells, and mononuclear Hodgkin cells, which are seen in specific cHL subtypes such as nodular sclerosis and mixed cellularity (Figure 1) [1,2]. In contrast, NLPHL lacks RS cells and instead features lymphocyte-predominant (LP) or “popcorn” cells, which display multilobulated, folded nuclei with smaller nucleoli. LP cells are positive for CD20, CD45, EMA, BCL6, and other B cell markers, reflecting their preserved B cell phenotype [3]. Morphologically, NLPHL shows a nodular architecture with a background rich in small B lymphocytes and follicular dendritic cells, differing from the mixed inflammatory milieu and architectural patterns seen in cHL subtypes [1,4]. cHL is a rare type of hematological cancer and has good prognosis; cHL accounts for about 95% of all cases of Hodgkin lymphoma (HL). HL accounted for 0.4% of new cancer cases and 0.2% of cancer-related deaths worldwide in 2020. In the United States, the disease had an estimated incidence rate of 0.26 cases per 100,000 people that year, representing 10% of all lymphoma diagnoses. cHL exhibits a bimodal age distribution, with peak incidence observed in young adults and those aged 55 and older [5,6,7,8].

The precise etiology of cHL remains unclear. However, several studies have identified an elevated risk in individuals with Epstein–Barr virus (EBV) infection. In resource-rich countries, EBV positivity in cHL ranges from 20% to 50%, with higher prevalence observed among older adults, children, and males. In contrast, resource-poor countries exhibit significantly higher EBV positivity rates, with the highest prevalence in Africa (74%) and lower rates in Europe (36%) and North America (32%), highlighting geographic differences in EBV-related HL cases [9,10]. This association is particularly notable in mixed cellularity and lymphocyte depletion subtypes, which is more common in the lower human development index and certain ethnic groups, even after accounting for socioeconomic factors [11,12].

EBV plays a significant role in cHL pathogenesis, with evidence suggesting it acts as an early pathogenic event. Monoclonal EBV genomes found in Hodgkin and Reed–Sternberg (HRS) cells indicate infection precedes clonal expansion. The virus employs type II latency, expressing proteins such as EBNA-1, LMP-1, and LMP-2, which promote tumorigenesis and create an immunosuppressive tumor microenvironment (TME) through cytokine induction and PD-L1 upregulation (Figure 2). This viral involvement is further highlighted in HIV-associated cHL, where nearly all cases are EBV-positive. Similarly, human immunodeficiency virus (HIV) infection increases the risk of cHL by 6–15 fold [9], alongside autoimmune disorders and immunosuppressive conditions. Moreover, familial predisposition has been identified as a contributing factor in Hodgkin lymphoma [1,13,14,15,16,17,18].

Additionally, cHL arises from complex genetic alterations primarily involving HRS cells, characterized by mutations in key driver genes affecting signaling pathways such as NF-κB and JAK/STAT. Additional genetic aberrations include alterations in SOCS1, PTPN1, TNFAIP3, and B2M, contributing to immune escape and cytokine dysregulation [19,20,21,22].

The recently updated 2023 WHO classification has introduced refined diagnostic criteria for immune deficiency/dysregulation-associated lymphoproliferative disorders (IDD-LPDs), integrating histologic subtype (hyperplasia, polymorphic disorder, or lymphoma), viral associations (particularly EBV latency states), and underlying immune dysfunction contexts (e.g., primary immunodeficiency, post-transplant, HIV). This standardized diagnostic framework, as proposed by the WHO, underscores the importance of accurately characterizing the immune environment and EBV latency patterns (latency I or II), thus enabling improved prognostication, targeted therapeutic strategies, and a deeper understanding of the interplay between genetic alterations, viral pathogenesis, and immune dysregulation in cHL (Table 1) [23,24].

The scope of this paper is to explore the potential causal link between viral infections, specifically EBV infection, and the development of cHL. It will examine the molecular mechanisms underlying viral contributions to cHL, including immune evasion, oncogenic signaling, and tumor microenvironment modulation, while addressing epidemiological disparities. The objective is to synthesize current evidence, highlight unresolved questions, and propose future research directions to deepen the understanding of viral-associated cHL and improve strategies for diagnosis and treatment.

## 2. Methodology

This narrative review aims to explore the potential causal link between viral infections, specifically EBV, HIV, hepatitis viruses, and the development of cHL. The review synthesizes findings from a wide range of primary studies, reviews, and expert opinions to provide a comprehensive understanding of the role of EBV, HIV, HCV, HBV, and their contribution in cHL pathogenesis.

The relevant literature was identified through searches of academic databases, including PubMed, Scopus, and Google Scholar, using search terms such as “classical Hodgkin lymphoma”, “viral oncogenesis”, “HCV AND HBV”, “Human herpesvirus”, “Sars-Cov-2”, “Covid-19”, “EBV and Hodgkin lymphoma”, “immunodeficiency”, “PTLD”, “immunosuppression”, and “HIV-associated lymphoma”. References from key articles were also reviewed to identify additional sources of relevance. The focus was placed on studies published in English that discuss viral involvement in cHL, its epidemiology, molecular mechanisms, immune modulation, and implications for treatment. The exclusion criteria omitted studies with limited data on viral mechanisms, non-peer review, non-English studies, or unrelated to cHL pathogenesis.

## 3. Causal Link Between EBV Viral Infection and cHL Development

### 3.1. Background on EBV and HL

EBV has long been suspected of playing a role in HL, supported by elevated antibody titers against EBV proteins in some patients. Research in the 1970s by IARC showed that EBV infection is nearly universal, with over 90% of adults worldwide carrying the virus. The presence of EBV in every tumor cell of EBV-positive cHL strongly supports its role in the disease’s pathogenesis. In Western populations, EBV is found in about one-third of cHL cases, with higher prevalence in other regions. If EBV were incidental, its frequency in cHL would reflect the rare occurrence of EBV-infected B cells (1 in 10^6^).

However, since cHL arises from GC B cells, the observed frequency aligns with the higher rate of EBV infection in GC B cells (4.6 per 10^4^) [25], reinforcing its etiological significance. EBV normally infects naive B cells in the Waldeyer ring, where it can differentiate into memory cells and exit the cell cycle without causing harm [26]. Over a 54-year period in the Brazilian state of São Paulo, the frequency of EBV-associated cHL showed a significant decline. Among the 817 cases studied, EBV positivity dropped from 87% in the period 1954–1979 to 46% in 2000–2008. This decline was most pronounced in children, adolescents, and young adults, while the rate remained relatively stable in patients over 45 years. These trends reflect a shift toward the epidemiological pattern seen in developed countries, likely influenced by socioeconomic improvements, which are associated with reduced early EBV exposure and enhanced immune control over the virus [27]. However, if EBV infects other types of cells, they may express the growth program and continue proliferating, a rare event that is usually controlled by cytotoxic T lymphocytes (CTLs). If the CTL response is suppressed, these cells can develop into post-transplant lymphoproliferative disorder (PTLD) or HIV-associated lymphomas [28,29].

PTLDs represent a group of lymphoid proliferations occurring due to therapeutic immunosuppression administered to prevent graft rejection after solid organ or hematopoietic stem cell transplantation [30]. The degree, duration, and type of immunosuppression significantly influence the development of PTLD, with potent immunosuppressants such as T-cell-depleting antibodies and calcineurin inhibitors associated with higher risks [29,31,32]. Immunosuppression impairs EBV-specific T-cell immune surveillance, enabling EBV-infected B cells to proliferate unchecked, particularly in patients who are EBV-seronegative prior to transplantation [33]. This immunologic dysfunction leads to a microenvironment enriched with exhausted and ineffective CD8+ T-cells overexpressing inhibitory markers like PD-1, which further supports malignant transformation and the progression of EBV-driven lymphoproliferative disease [29,32,34].

Hodgkin lymphoma arises from the default program (Latency II) of EBV infection, while Burkitt lymphoma develops when a germinal center cell entering the memory compartment becomes stuck in proliferation, expressing only EBNA-1 [17,35,36,37,38]. However, given that over 90% of humans harbor latent EBV infections, initial findings were inconclusive. Significant advances occurred in 1985 and 1989 when the EBNA protein (Epstein–Barr nuclear antigen, now known to be EBNA1) was detected in HRS cells, indicating clonal viral infection and suggesting that HRS cells are monoclonal [39,40,41,42].

### 3.2. EBV-Mediated Oncogenesis

EBV plays a critical role in the pathogenesis of certain lymphomas, including cHL, Burkitt lymphoma (BL), and diffuse large B cell lymphoma (DLBCL) [43,44] (Table 2). EBV infection can be lytic, involving widespread viral gene expression, virus production, and cell death [45], or latent, observed in most tumor cells, with limited gene expression, persistent infection, and host cell transformation. EBV infects HRS cells clonally, promoting their survival by preventing apoptosis and upregulating PDL1 expression, mediated by latent membrane protein 1 (LMP1), which is a viral protein [14].

HRS cells express EBNA1, LMP1, LMP2A, and LMP2B, while the other EBNAs are downregulated [46], HRS cells bypass normal GC B cell survival mechanisms, relying on nonphysiological pathways. LMP1 acts as an oncogene by mimicking CD40 signaling [47,48], activating AP-1, JAK\STAT, NF-κB, Jun N-terminal kinase (JNK), and p38 mitogen-activated protein kinase and other pathways through its cytoplasmic domain [49,50,51,52,53,54].

The NF-κB pathway is a cornerstone of HRS cell survival, driving the upregulation of anti-apoptotic genes that protect these malignant cells from programmed cell death [55,56,57]. LMP1 further amplifies this effect in GC B cells by inducing Id2 expression [58], which suppresses the E2A transcription factor and silences B cell signature genes, facilitating extensive transcriptional reprogramming [58]. Beyond its role in cellular survival, NF-κB activation orchestrates a broad spectrum of critical processes, including immune system regulation, cell proliferation, viral replication, tumor metastasis, and inflammation [56,57,59]. This multifaceted activity positions NF-κB as a central player in tumor progression, immune evasion, and the pathogenesis of cHL; however, these pathways may also be activated through genetic mutations within the cells [56,57,59].

Meanwhile, LMP2a resembles components of the B cell receptor (BCR) and sustains survival signaling [48], LMP2A contributes to HRS cell survival and transcriptional reprogramming by activating Notch1, inhibiting E2A, and downregulating EBF, key transcription factors for B cell development [26,60,61]. However, the exact role of LMP2A in cHL remains unclear due to the downregulation of many BCR signaling molecules in HRS cells, although its effects extend beyond classical BCR pathways. Furthermore, latently infected cells transcribe noncoding viral RNAs, including two small non-polyadenylated transcripts called EBERs and more than 44 viral microRNAs (miRNAs). These miRNAs are either embedded within the introns of BARTs or located near the coding region of the BHRF1 gene; in cHL, cells express BART miRNAs but lack BHRF1 miRNAs which is linked to latency type III. The simultaneous expression of all latent genes characterizes what is known as latency III [36,37,38,40,62,63,64,65].

Certain miRNAs such as BART2-5p are expressed in cHL and help suppress the lytic cycle by downregulating the viral DNA polymerase BALF5, supporting the maintenance of latency [66,67]. BART2-5p can also impair B cell receptor signaling. By aligning with the characteristic loss of BCR function in cHL, it can also inhibit the expression of the viral DNA polymerase BALF5, thereby suppressing the lytic cycle and promoting latency [66,68]. Additionally, BART13-3p is highly expressed and secreted via exosomes, where it can alter macrophage behavior to promote a tumor-supportive microenvironment by increasing cytokines like TNF-α and IL-10 [69,70]. The expression patterns of EBV miRNAs also shift across stages of latency, with different subsets activated during transitions from latency III to latency II and 0 [71]. However, detecting EBV miRNAs in cHL is challenging due to the low number of malignant HRS cells and contamination by surrounding non-neoplastic immune cells [72]. Additionally, EBV can influence host miRNA expression; in EBV-positive cHL, miR-96, miR-128a, and miR-128b are notably downregulated [73]. These findings underline the multifaceted role of EBV miRNAs in shaping the TME, maintaining latency, and modulating immune escape mechanisms in cHL. Recent findings emphasize the importance of both host and EBV-encoded microRNAs as potential circulating biomarkers in EBV-associated lymphomas, including cHL [74,75,76]. While EBV miRNAs such as BART13-3p have been detected in exosomes and linked to immune modulation, several host miRNAs (e.g., miR-21, miR-155) are also differentially expressed and could serve as minimally invasive tools for diagnosis or prognosis [70,72].

### 3.3. Differences Between EBV-Positive and EBV-Negative cHL

EBV-positive cHL is typically associated with MCHL and LDHL and shows a dense inflammatory background composed of numerous CD68+ and CD163+ macrophages, cytotoxic T lymphocytes, and frequent necrotic areas [35,77]. The RS cells are often large, with prominent nucleoli, and test positive for EBV markers such as LMP1 and EBER [78]. In contrast, EBV-negative cHL, commonly seen in nodular sclerosis and lymphocyte-rich subtypes, displays a more fibrotic architecture with lacunar-type HRS cells, fewer macrophages, and a less pronounced inflammatory infiltrate, reflecting fundamental histopathological differences between the two variants [13].

In EBV-negative cHL, oncogenesis is driven by mutations in genes like C-REL and tyrosine kinase receptors [79,80,81,82]. Additionally, hypothetical cause of cHL in EBV-negative cases is alternative pathogenic mechanisms unrelated to viral influence, which include genetic abnormalities, aberrant signaling pathways, and interactions with the TME. Specifically, EBV-negative cHL often exhibits dysregulation of immune checkpoints, such as PD-1/PD-L1, and a distinct inflammatory microenvironment characterized by a Th17 profile, which differs from the Th1-dominant environment in EBV-positive cases [13,44,83,84].

EBV-infected HRS cells also foster an immunosuppressive environment by interacting with immune cells and secreting suppressive mediators [85,86]. EBV causes the overexpression of PD-L1, PD-L2, and CTLA-4 [87,88], further amplifying immune evasion mechanisms through inhibiting T-cell activation. PD-1 (CD279) inhibits T-cell activity upon engagement with its ligands PD-L1 and PD-L2. CTLA-4 (CD152) competes with CD28 for binding to CD80/CD86, suppressing T-cell activation, leading to immune exhaustion or apoptosis. T-cell activation depends on TCR signaling and co-stimulatory signals, regulated by immune checkpoints like PD-1 and CTLA-4, which maintain immune tolerance and prevent overactivation. In EBV-positive lymphomas, the virus enhances immune evasion by upregulating PD-L1 (Figure 3) and PD-L2 through the NF-κB and JAK/STAT pathways, with additional contributions from 9p24.1 genetic alterations. These mechanisms impair T-cell function, leading to immune suppression and facilitating lymphoma progression. Additionally, EBV-positive HL is linked to the suppression of the gene p21^cip1/waf1, which plays a key role in cell cycle regulation and apoptosis. The EBV-encoded RNA EBER1 was shown to inhibit p21^cip1/waf1 transcription by downregulating its regulators, p53, EGR1, and STAT1, making cells more resistant to apoptosis induced by common lymphoma treatments. Clinically, EBV+ HL cases exhibited lower p21^cip1/waf1 expression and had significantly worse two-year survival outcomes compared to EBV- cases, suggesting that EBER1′s anti-apoptotic effect contributes to treatment resistance and poorer prognosis [13,40,64,65,86].

Studies have demonstrated that immune checkpoint blockade therapies can effectively restore T-cell activity, reduce tumor burden, and counteract the immune evasion caused by EBV-driven PD-L1 expression. The expression occurs through the activation of the AP1 transcription factor pathway, which upregulates c-Jun and JunB. Notably, PD-L1 expression is observed in 89.2% of HRS cells and extensively in the TME of cHL patients. Despite this high prevalence, studies have shown no significant correlation between PD-L1 expression, EBV status, clinical parameters, or prognosis. This pathway, a hallmark of cHL, functions independently of 9p24.1 copy number variations, underscoring its potential as a target for emerging immunotherapies. EBV’s latency states influence its role in lymphomas. In BL and related B cell malignancies, EBV employs latency type 1, expressing EBNA1, EBERs, and BARF0 to evade immune detection. In cHL and NK/T-cell lymphomas, latency type 2 predominates, involving LMP1, BARF0, EBERs, EBNA-1, and LMP2 expression but not EBNA2. Latency type 2 is a target for PD1/PDL1-blocking therapies. In immunosuppressed individuals, latency type 3, expressing a broader range of EBV proteins, emerges. EBV-positive and EBV-negative tumors exhibit distinct pathogenic mechanisms and prognoses [13,35,40,64,65,86,89,90,91,92,93].

In EBV-positive cHL cases, LMP1 and LMP2a provide critical survival signals to GC B cells, compensating for the absence of functional BCRs due to destructive IgV mutations. Notably, EBV-positive HL cases often harbor fewer mutations in genes regulating the NF-κB pathway, such as TNFAIP3 and NFKBIA, due to the strong NF-κB activation. Additionally, EBV-positive cases show lower overall mutation loads in oncogenes and tumor suppressor genes compared to EBV-negative cases, suggesting a pathogenetic role for EBV; viral genes alone can drive abnormal signaling pathway activation, reducing the need for cellular mutations. This suggests that EBV and cellular mutations represent distinct, mutually exclusive mechanisms leading to the same disease outcome. Beyond protein expression, EBV-infected cells produce noncoding RNAs, such as miR-BHRF1-2-5p and miR-BARTs. These ncRNAs target and modulate PD-L1/PD-L2 expression, further contributing to immune suppression. Collectively, these findings underscore EBV’s significant role in the survival and transformation of pre-apoptotic GC B cells, particularly in EBV-positive HL cases [37,40,89,94].

## 4. Genetic Associations

A study in 2015 examined HLA associations with EBV-positive and EBV-negative cHL using allele selection modeling. For EBV-positive cHL, HLA-A01:01 and B37:01 were associated with increased risk, while DRB115:01, E*01:01 Is, and DPB101:01 were linked to decreased risk. In EBV-negative cHL, the strongest predictor was SNP rs6903608 in the class II region, with B15:01, DRB103:01, and DQB1*03:03 associated with increased risk. These findings reveal distinct HLA associations for EBV-positive and EBV-negative cHL, suggesting different mechanisms in disease pathogenesis and highlighting the importance of analyzing EBV-stratified subgroups in cHL research [95,96,97].

Additionally, a genetic variant, rs6457715, located near the HLA-DPB1 gene in the MHC region, was identified as strongly associated with EBV-positive cHL. This variant confers a 2.3-fold increased risk specifically for EBV-positive cHL, with no significant association for EBV-negative cHL. The effect is consistent across histological subtypes, indicating that EBV status is the primary driver of the association. rs6457715 represents the third independent genetic locus in the MHC region exclusively associated with EBV-positive cHL. These findings reinforce the critical role of EBV in cHL etiology and the importance of considering tumor EBV status in genetic susceptibility studies [98,99,100].

In summary, EBV plays a significant causal role in cHL, particularly in immune-suppressed settings, where it provides survival signals to B cells through latency proteins like LMP1 and LMP2a. EBV-positive cHL cases rely less on cellular mutations for NF-κB activation due to strong viral signaling, while EBV-negative cases involve alternative mechanisms, including genetic mutations, socioeconomic interactions, and distinct HLA associations. These findings highlight the importance of EBV status in understanding cHL pathogenesis and its implications for targeted therapies.

## 5. Beyond EBV: HIV Infections in CHL Development

### 5.1. Background

Classic HL is the most prevalent non-AIDS defining tumor. HIV-positive individuals have a significantly elevated risk of developing cHL, with incidence rates 5–26 times higher than those observed in the general population [101]. This increased vulnerability is attributed to the combined effects of HIV-induced immune suppression and frequent co-infection with EBV. Studies show that nearly 90–100% of cHL cases in HIV-infected individuals are EBV-positive, compared to only 30–40% in HIV-negative cases, highlighting the virus’s central role in lymphomagenesis. HIV-related cHL predominantly presents as the mixed cellularity subtype, followed by nodular sclerosis. Pathologically, it shares similarities with cHL in HIV-negative individuals, characterized by HRS cells in a background of lymphocytes, eosinophils, neutrophils, macrophages, and plasma cells. However, HIV-related cHL features a higher number of HRS cells, frequent co-infection with EBV, a “sarcomatoid” pattern, and distinct microenvironmental changes, including reduced CD4+ T-cell counts, an inverted CD4/CD8 T-cell ratio, a reduction in infiltrating GrB+ cells (activated cytotoxic cells), and an increase in infiltrating TIA+ T-cells (primarily non-activated cytotoxic cells) were noted.

Adding to this complexity, a recent genetic study has revealed significant heterogeneity in HIV-associated lymphomas. Next-generation sequencing of diagnostic biopsies has identified 213 mutations across 42 genes, with frequent mutations in KMT2D (67%), TP53 (61%), and TNFAIP3 (50%). Most mutations are unique to individual patients, reflecting diverse genetic and pathophysiological mechanisms. This genetic diversity underscores the multifaceted drivers of lymphomagenesis in HIV-positive individuals, further compounded by the immune dysregulation caused by HIV.

Chronic antigenic stimulation by HIV leads to B cell hyperactivation, increasing the likelihood of genetic mutations and chromosomal translocations that predispose cells to malignant transformation. Furthermore, the depletion of CD4+ T-cells impairs the immune system’s ability to surveil and eliminate aberrant cells. Interestingly, a moderately suppressed immune state appears more conducive to cHL development than severe immunosuppression, as some level of immune activity is necessary to create the pro-inflammatory microenvironment that supports the growth and survival of HRS cells [93,101,102,103,104,105,106,107].

In summary, the development of cHL in HIV-positive individuals results from a complex interplay between immune suppression, EBV-driven oncogenesis, and chronic immune activation. These factors collectively create a permissive environment for the transformation and proliferation of malignant B cells, contributing to the disproportionately high incidence of cHL in this population.

### 5.2. TME in HIV-Associated and EBV-Positive cHL

TME in HIV-associated cHL is shaped by a complex interplay of HIV-induced immune suppression and EBV-driven oncogenesis. HIV-associated cHL predominantly features unfavorable histological subtypes, such as mixed cellularity and lymphocyte-depleted cHL, characterized by extensive necrosis and sarcomatoid macrophage infiltration. EBV universally infects HRS cells in this context, with its LMP1 driving NF-κB activation, chronic inflammation, immune evasion, and tumor cell survival. Compared to non-HIV cHL, the TME in HIV-associated cHL is enriched with CD68+ and CD163+ spindle-shaped M2 macrophages, PD-L1-expressing cells, and Galectin-1 and significantly lower levels of CD4+ T-cells, CD56+ NK cells, CD57+ cells, CD123+ dendritic cells, and B cells in the TME of HIV-positive cHL cases compared to HIV-negative ones, fostering a highly immunosuppressive environment that shields HRS cells from immune attack [101,108,109,110,111].

HIV/EBV co-infection further alters the TME by disrupting immune dynamics. While EBV-positive cHL typically displays an inflammatory profile with increased CD8+ T-cell activity, HIV co-infection reduces CD8+ T-cell density and functionality, downregulating key T-cell receptor (TCR) signaling mediators such as VAV1, FYN, and AKT1. Additionally, the spatial organization of effector cells is disrupted, with CD8+ T-cells and GrzB+ T-cells found farther from neoplastic cells. These changes are accompanied by extracellular matrix (ECM) remodeling, including disrupted CXCL13 networks and reduced heparan sulfate levels, impairing T-cell trafficking and antigen presentation. The immune evasion mechanisms are further enhanced by the increased expression of coinhibitory ligands like CD155 and nectin-3, promoting interactions with inhibitory receptors on T-cells, such as TIGIT [112,113].

These dynamic interactions create a permissive environment for tumor progression and immune escape. Immune checkpoint inhibitors, such as pembrolizumab and nivolumab, show promise in reactivating suppressed immune responses, while ART may partially reduce inflammation and regulatory T-cell activity. However, its effects on TME immune activation remain incomplete, underscoring the need for combinatorial therapies targeting these specific molecular alterations in HIV/EBV-associated cHL [108,112,113].

### 5.3. Management

The treatment of EBV-positive cHL and HIV-associated cHL in immunosuppressed individuals aligns with protocols for the general population but requires the integration of combination antiretroviral therapy (cART) or reducing immunosuppression. However, HIV-positive individuals on cART have a high risk of developing HL, particularly in the early months of therapy [23]. HIV-HL presents with a more aggressive phenotype, advanced-stage disease, and atypical tumor spread. Most patients have moderate immunosuppression, and a declining CD4 count can be an early HL marker [106,107,108].

Historically, outcomes were poor, but the combination of cART, chemotherapy, and improved supportive care has significantly enhanced survival [114]. However, disparities in treatment access continue to impact prognosis, emphasizing the need for broader implementation of effective management strategies [115].

For HIV-related cHL, standard treatment includes ABVD (doxorubicin, bleomycin, vinblastine, dacarbazine) combined with cART, achieving complete response (CR) rates of 74% and 5-year overall survival (OS) of 81%. In limited-stage HIV–HL, the recommended approach is two cycles of ABVD followed by 20 Gy involved-site radiation therapy (ISRT) [114,116]. Data support using lower radiation doses (20 Gy instead of 30 Gy) and smaller ISRT fields because of good disease control and reduced toxicity [116]. BEACOPP (bleomycin, etoposide, doxorubicin, cyclophosphamide, vincristine, procarbazine) is an alternative for advanced stages with consolidation radiation therapy reserved for FDG-PET-CT-positive residual disease but is limited by toxicity; brentuximab vedotin (BV)-AVD may be considered for stage IV disease in select cases despite limited data [116]. Interim PET-CT guides risk-adapted therapy, with negative scans predicting improved progression-free survival (PFS) but this still needs confirmation [116]. Novel therapies like brentuximab vedotin (anti-CD30 antibody-drug conjugate) combined with AVD achieved a 2-year PFS of 86% and a 2-year overall survival rate of 92%, with manageable toxicity [9]. Immune checkpoint inhibitors (ICIs) have transformed cancer treatment, including cHL, with pembrolizumab and nivolumab showing high efficacy (65–87% response rates) in relapsed/refractory HIV-negative cHL. However, data on ICIs in HIV-associated cHL are limited, as PLWH have been excluded from most trials. A phase I study found pembrolizumab safe in PLWH with advanced cancers, warranting further research [9,117]. Only two trials confirm ICIs’ safety in PLWH, with no impact on viral suppression or CD4 counts [23].

For relapsed/refractory cases, high-dose chemotherapy (ESHAP, DHAP, ICE) followed by autologous stem cell transplantation (ASCT) is the standard [118,119], with real-world data indicating a higher infection risk in lymphoma patients with HIV compared to those without, emphasizing the need for infection prevention and supportive management [23]. Chemotherapy with cART poses drug interaction risks, particularly with CYP3A4 inhibitors, though integrase inhibitors offer safer alternatives. In EBV-associated PTLD, rituximab with chemotherapy and EBV-specific cytotoxic T-cell (EBV-CTL) therapy are effective with overall response rates of 68% following HSCT and 54% after SOT, and no significant toxicity observed [29,32,120].

Those who relapse post-ASCT can receive PD-1 inhibitors such as nivolumab or pembrolizumab. Patients ineligible for HDCT should follow HIV-negative HL recommendations, with a preference for pembrolizumab given data from a prospective phase III trial [23,106,107,108,116,120,121,122,123].

Recent evidence highlights the significance of LAG-3 as an immune checkpoint in cHL, particularly in the context of EBV-associated cases. A study by Jimenez et al. demonstrated that LAG-3 was expressed in 57% of pediatric cHL cases, with 70% of these being EBV-positive. Notably, a strong correlation between LAG-3 and PD-1 expression was observed exclusively in EBV+ cases, suggesting that EBV may promote a synergistic immune exhaustion phenotype [124]. This dual expression was associated with significantly worse event-free survival (54% vs. 100% in other subgroups), underscoring the potential role of LAG-3 and PD-1 in fostering a tolerogenic microenvironment that facilitates lymphomagenesis [124]. These findings support the rationale for dual checkpoint inhibition strategies targeting both PD-1 and LAG-3 in EBV+ cHL, an approach currently being explored in clinical trials. More recent clinical data further supports the therapeutic relevance of targeting LAG-3 in cHL, particularly in patients with relapsed R/R disease following anti-PD-1 therapy. In an open-label phase 1/2 trial (NCT03598608), the combination of favezelimab (an anti-LAG-3 antibody) and pembrolizumab demonstrated promising antitumor activity in heavily pretreated R/R cHL patients who had progressed after prior PD-1 blockade. The objective response rate was 31%, with 79% achieving some degree of tumor reduction. Notably, dual LAG-3/PD-1 inhibition showed a tolerable safety profile and yielded a 12-month PFS rate of 39%, with a 91% OS rate. These findings underscore the potential of LAG-3/PD-1 co-blockade to re-induce responses in patients who have otherwise exhausted standard immunotherapeutic options, reinforcing the importance of incorporating LAG-3 in the immune checkpoint landscape of cHL [125].

Summing up, both HIV-associated and EBV-driven lymphomas treatment combine the control of the underlying immunosuppression and targeted oncologic therapy (Table 3) to improve survival and reduce relapses. In HIV-positive cases, cART is continued throughout chemotherapy to restore immune function, maintain virologic suppression, and diminish opportunistic infections. HL in immunodeficient populations, typically EBV-positive, generally responds to front-line ABVD chemotherapy, sometimes adapted by stage or combined with risk-based radiotherapy. In relapsed disease, high-dose chemotherapy and autologous stem cell transplantation can be considered, and agents such as brentuximab vedotin and ICIs may offer benefits for refractory cases; however, the efficacy of ICIs requires further validation through larger cohorts and additional clinical trials. PTLD involving EBV often regress with immunosuppression reduction, but persistent or polymorphic lesions may require rituximab or chemoimmunotherapy, similar to HIV lymphomas but adjusted for transplant organ function. Across all these settings, reconstitution or modulation of immune status is fundamental, cART in HIV, immunosuppression tapering in transplant recipients, or novel T-cell-directed strategies, because restoring or optimizing antiviral immune surveillance synergizes with standard lymphoma-directed regimens to achieve better control of these virally driven malignancies. Additionally, novel EBV-targeted therapies focus on neutralizing or disrupting tumor-promoting miRNAs. Tools such as miRNA sponges, anti-miRNA oligonucleotides, and CRISPR/Cas9 are rapidly advancing, offering promising, more precise strategies to counteract EBV-driven malignancies by specifically blocking or editing viral miRNAs [23,29,32,106,107,108,116,120,121,122,123,126] (Table 3).

### 5.4. Preventive Measures

In people living with HIV (PLWH), EBV plays a central role in the pathogenesis of lymphoproliferative disorders, including cHL, which is frequently EBV-positive (up to 90% in HIV-associated cases). Chronic immune dysregulation caused by HIV, marked by CD4+ T-cell depletion, cytokine dysregulation, and persistent inflammation, creates an environment permissive for EBV reactivation and lymphomagenesis. The article by [23], 2024 underscores that initiating cART early in PLWH reduces HIV viremia, restores immune function, and lowers the risk of EBV-driven malignancies by mitigating immune activation and improving viral control. Persistent HIV replication (≥100,000 copies/mL) and low CD4 counts (<50/μL) correlate with heightened lymphoma risk, emphasizing the need for strict cART adherence. While no direct EBV-targeted prevention exists for PLWH, lifestyle modifications (e.g., smoking cessation), vaccinations (e.g., HBV, HPV), and avoidance of co-infections (e.g., KSHV/HHV8) are recommended to minimize immune dysregulation and secondary cancer risks. The use of colony-stimulating factors like G-CSF after each chemotherapy cycle is strongly recommended for primary infection prophylaxis. Additionally, all PLWH undergoing chemotherapy or radiotherapy should receive prophylaxis against *Pneumocystis jirovecii*, as these treatments reduce CD4+ lymphocyte counts. Cotrimoxazole is the preferred option, offering added protection against bacterial infections and toxoplasmosis. For patients with CD4+ counts below 50/µL, oral azithromycin is advised to prevent *Mycobacterium avium* complex infections. The article also notes that EBV-positive cHL in PLWH often presents with advanced-stage disease and extranodal involvement, necessitating extensive monitoring for lymphoproliferative symptoms in this population [23,106,130].

## 6. Beyond EBV: Other Viruses

A recent 2021 study evaluated the association between hepatitis B virus (HBV) and hepatitis C virus (HCV) infections and the development of HL. A systematic review and meta-analysis found inconclusive evidence for a link between these hepatitis infections and HL. For HBV, pooled data suggested a relative risk (RR) of 1.39, but this was not statistically significant. Similarly, HCV studies showed an RR of 1.09, also lacking significance. Overall, while some patterns suggest a possible association between hepatitis infections and HL under specific conditions, the evidence remains insufficient to establish a direct causative link. Further research is needed to explore these relationships in diverse populations, particularly considering factors like immune status and Epstein–Barr virus co-infection [131].

A study from 2019 investigated Human herpesvirus-6 (HHV-6) for its potential role in HL using IARC criteria for oncogenicity, but the evidence remains inconclusive. Studies have detected HHV-6 DNA in some HL cases, but it is not consistently present in all tumor cells, failing a key criterion for establishing causality. Elevated HHV-6 antibody levels have been reported in some HL patients, but these findings are inconsistent across studies. Unlike EBV, which is firmly linked to HL, HHV-6′s association with the disease appears limited or secondary. Current research suggests that HHV-6 may contribute indirectly or be a passenger virus rather than a direct cause of HL [132].

## 7. Challenges

The heterogeneity of cHL reflects significant challenges and controversies, especially regarding EBV prevalence and impact. EBV positivity varies widely across cHL subtypes and populations. For instance, in developing regions, mixed cellularity subtype is frequently EBV-positive, particularly in children and older adults, while the nodular sclerosis subtype, which is more common in industrialized nations, tends to be EBV-negative. Further complicating the picture is the variability in EBV protein expression. Proteins like LMP1 and LMP2A, expressed in EBV-positive cases, contribute to immune evasion and oncogenesis by promoting NF-κB activation and altering the TME. However, this pathogenic impact differs among cases, influenced by genetic predispositions, immune responses, and geographic or socioeconomic factors. These disparities not only challenge uniform diagnostic and therapeutic strategies but also highlight the complex interplay of viral, host, and environmental factors in cHL [14,27].

Establishing causality in viral associations with cHL presents significant challenges due to the lack of direct evidence linking some viruses to the disease. While EBV is frequently detected in tumor cells, the precise mechanisms by which it contributes to oncogenesis remain incompletely understood. Confounding factors, such as genetic susceptibility and environmental exposures, further complicate this relationship. Genetic predispositions, including variations in HLA alleles, may influence immune responses to EBV, while environmental conditions, such as socioeconomic status and early-life infections, can alter disease patterns, making it difficult to isolate the virus’s role in cHL pathogenesis.

## 8. Future Directions

Future research on immune deficiency/dysregulation-associated EBV-positive cHL should focus on integrating large-scale, longitudinal, and collaborative epidemiological efforts to clarify incidence, patterns of immunosuppression, and viral cofactors across diverse geographical and socioeconomic settings. Comprehensive genomic and environmental data, coupled with HLA typing and immune profiling, can illuminate gene or environment interactions and refine risk stratification. Advances in virus-targeted therapeutics, ranging from EBV-specific T-cell infusions and CRISPR/Cas9-based strategies to prophylactic or therapeutic EBV vaccines, hold promise for more precise interventions. Recent studies have successfully identified and characterized the first functional HLA-A01:01-restricted T-cell receptors (TCRs) specific for EBV LMP2, demonstrating their ability to recognize and kill target cells in vitro. These TCRs, when engineered into CD8+ and CD4+ T-cells, restored cytotoxic activity against EBV-infected cells and may serve as a foundation for adoptive T-cell therapies in EBV-driven lymphomas, including cHL. This approach highlights the potential of precision immunotherapy that directly targets viral antigens involved in lymphomagenesis [133].

Emerging classification frameworks that incorporate histopathologic features, underlying immune context, and advanced molecular diagnostics will facilitate more accurate diagnosis and prognostication, especially in complex settings like HIV infection or post-transplant immunosuppression. Equitable access to these diagnostic and therapeutic advances depends on addressing global disparities in healthcare infrastructure, training, and policy, underscoring the importance of capacity-building programs and real-world implementation science to evaluate outcomes in heterogenous populations. A multi-disciplinary approach, often involving oncology, infectious disease, immunology, and transplant specialists, is critical for managing opportunistic infections, optimizing immune reconstitution (through antiretroviral therapy or immunosuppression tapering), and coordinating supportive care measures.

While EBV remains the principal pathogen of interest, further research into co-infection scenarios, such as hepatitis viruses, KSHV, or even SARS-CoV-2, could clarify additional mechanisms of viral-driven lymphomagenesis and pave the way for multi-targeted prophylaxis or intervention strategies. Ultimately, by embracing precision oncology, strengthening evidence through robust clinical trials, and ensuring global accessibility to advanced diagnostics and treatments, future efforts aim to transform EBV-positive cHL into a more preventable and precisely managed malignancy for both immunocompetent and immunocompromised populations worldwide.

## 9. Conclusions

cHL is a biologically diverse disease, heavily influenced by immune status and viral infections, notably EBV. In immunocompromised individuals, EBV serves as a key driver of lymphomagenesis through viral oncoproteins (LMP1 and LMP2A), which activate essential survival signaling pathways such as NF-κB, reshape the tumor microenvironment, and facilitate immune evasion. These molecular and clinical characteristics clearly distinguish EBV-positive from EBV-negative cHL, highlighting the necessity for revised diagnostic frameworks, as recently emphasized by the WHO 2023 classification.

Clinically, this distinction guides the implementation of targeted treatments, including immune checkpoint inhibitors, EBV-specific T-cell therapies, and emerging miRNA-based interventions, which offer the potential for improved therapeutic outcomes. Moving forward, concerted efforts in epidemiological research, molecular diagnostics, and precision medicine strategies, particularly in underserved populations, are critical. These steps will ensure better disease management and ultimately reduce disparities in clinical outcomes globally.

## Figures and Tables

**Figure 1 cancers-17-01433-f001:**
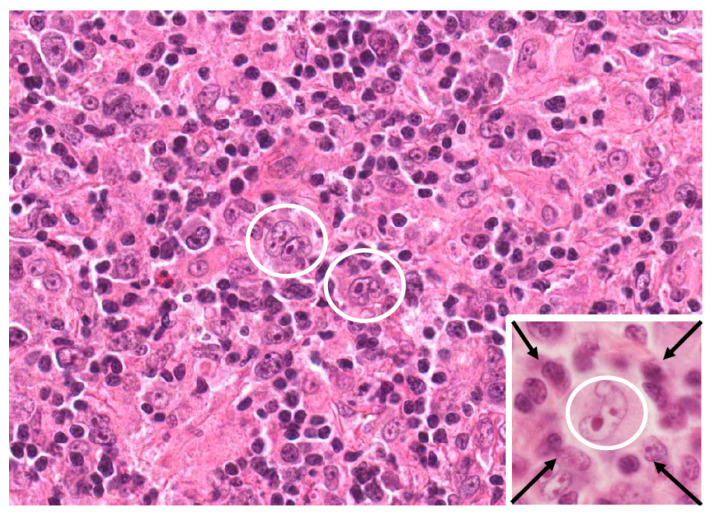
Immune deficiency/dysregulation-associated classic Hodgkin lymphoma (cHL) (HIV-related subset). EBV-positive cHL from an HIV-infected patient. Two Hodgkin–Reed–Sternberg (HRS) cells (within white circle) are recognizable as large multinucleated cells within inflammatory cellularity showing “mixed cellularity” pattern. In inset, HRS cell (within white circle) is “rosetted” by small lymphocytes (arrows). Haematoxliyn and Eosin stain; original magnification 40×, inset 100×.

**Figure 2 cancers-17-01433-f002:**
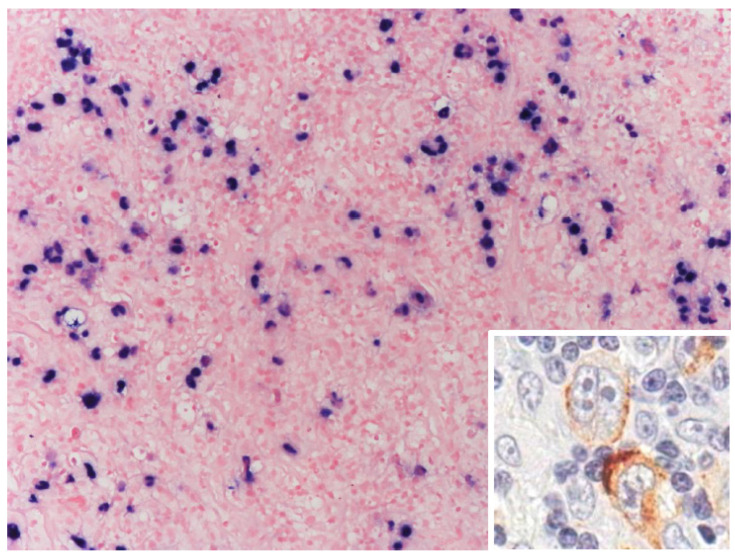
Immune deficiency/dysregulation-associated classic Hodgkin lymphoma (cHL) (HIV-related subset). EBV-positive cHL from HIV-infected patient. Hodgkin Reed–Sternberg cells (HRS) are EBV-positive, as demonstrated by in situ hybridization, for EBERs. Positivity is nuclear. Inset: EBV-infected HRS cells express, as demonstrated by immunohistochemistry, latent membrane protein 1. Staining is cytoplasmic. Original magnification 20×; inset: 100×.

**Figure 3 cancers-17-01433-f003:**
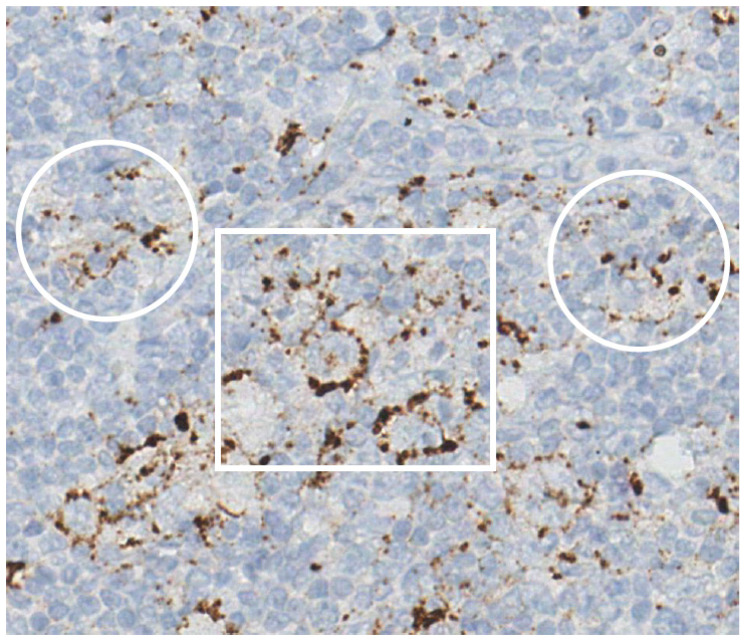
Immune deficiency/dysregulation-associated cHL (HIV-related subset). EBV-positive classic Hodgkin lymphoma (cHL) from HIV-infected patient. PD-L1 expression in Hodgkin Reed–Sternberg (HRS) cells (within white rectangle) and macrophages (areas highlighted by white circles) in cHL. Immunohistochemical staining; Haematoxylin counterstain; original magnification 40×.

**Table 1 cancers-17-01433-t001:** Morphologic characteristics and EBV infection in immune deficiency/dysregulation-associated classic Hodgkin lymphoma.

Immunodeficiency-Associated cHL	Hodgkin Lymphoma Subtype	EBV Infection
Immune senescence-associated cHL	cHL, nodular sclerosis	Usually Neg
	cHL, mixed cellularity	Usually Pos
	Rare Types	
	cHL, lymphocyte rich	Variably Pos
	cHL, lymphocyte depleted	Variably Pos
HIV-associated cHL	cHL, nodular sclerosis	Pos
	cHL, lymphocyte depleted	Pos
	cHL, mixed cellularity	Pos
	cHL, lymphohistiocyoid	Pos
Post-transplant (cHL type PTLD)	Similar subtypes as above	Pos
Other iatrogenic immune deficiency-associated cHL	cHL, mixed cellularity	Usually Pos

Abbreviations: cHL, classic Hodgkin lymphoma; PTLD, post-transplant lymphoproliferative disorder; Neg, negative; Pos, positive.

**Table 2 cancers-17-01433-t002:** EBV-associated lymphomas.

Lymphoma Type	Approximate Percentage of EBV Positivity	EBV Latency
Burkitt lymphoma		
Endemic	>95	I
Sporadic	20–80	I
HIV-associated	30–50	I
HIV-associated DLBCL		
Immunoblastic	70–100	II/III
Non-immunoblastic	10–30	I
CNS lymphomas	>95	II/III
Plasmablastic	60–75	I/II
Primary effusion lymphoma	70–90	I
Post-transplant LPD	>90	II/III
EBV-associated DLBCL (of the elderly)	100	II
Classic Hodgkin lymphoma *	20–90	II
Extranodal T/NK cell lymphoma nasal type	100	II

* Classic HL of the general population include the following subtypes: nodular sclerosis (usually EBV-negative), mixed cellularity (usually EBV-positive), lymphocyte rich (variably EBV-positive), and lymphocyte depleted (variably EBV-positive). Association with EBV is less frequent in nodular sclerosis (10–40%) than in mixed cellularity cHL (approximately 75%). For immune deficiency/dysregulation-associated classic Hodgkin lymphoma, see Table 1. Abbreviations: CNS, central nervous system; DLBCL, diffuse large B cell lymphoma; lLPD, lymphoproliferative disorder.

**Table 3 cancers-17-01433-t003:** Current and novel therapies for immune deficiency/dysregulation-associated classic Hodgkin lymphoma.

Category	Treatment Strategy	Key Outcomes	Reference
General treatment approach	Follows general population protocols but integrates cART or reduces immunosuppression.	Survival improvement but disparities in treatment access persist.	[23,106,107,108,116,120,121,122,123]
HIV-HL standard treatment	ABVD + cART.	High response rates with ABVD; 74% CR rate, 5-year OS of 81%. Integration with cART essential.	[114,116]
Limited-stage HIV-HL	Two cycles of ABVD + 20 Gy ISRT; lower doses (20 Gy) and smaller fields reduce toxicity.	CR: 96%, 2-yr OS: 95.7%. Reduced radiation dose is effective; lower toxicity, good disease control.	[114,116]
Advanced-stage HIV-HL	ABVD/BEACOPP baseline; BV-AVD OR nivolumab-AVD for stage IV in select patients; PET-positive residual disease requires consolidation radiation.	ABVD/BEACOPP baseline: 2-yr PFS: 87.5%, 2-yr OS: 86.8%, 4 toxicity-related deaths with >6 BEACOPP cycles; max 6 recommended.Effective for aggressive disease; PET-CT guides therapy for improved outcomes.	[108,113,116,127]
Relapsed/refractory cases	High-dose chemotherapy (ESHAP, DHAP, ICE) followed by ASCT; higher infection risk in HIV patients.	Higher infection risk in HIV patients undergoing ASCT; supportive care is key.	[118,119]
Novel therapies	BV-AVD, ICIs (pembrolizumab, nivolumab) show high efficacy but need validation in HIV-HL.	BV-AVD: 2-year PFS 86%, OS 92%. Strong potential, but ICIs require larger trials in HIV-HL patients.	[9,117,120,128,129]
Chemotherapy and cART considerations	Chemotherapy with cART poses drug interaction risks; integrase inhibitors are safer; ICIs safe in PLWH with no impact on viral suppression.	Drug interactions manageable with proper selection of ART regimens.	[120]
PTLD management	Rituximab + chemotherapy or EBV-CTL for PTLD.	ORR of 68% post-HSCT, 54% post-SOT. Effective in PTLD; good response rates with acceptable toxicity. PTLD is treated like cHL but requires careful adjustments due to poorer tolerance, limited efficacy of immunosuppression reduction.	[29,32,120]
Emerging EBV-targeted therapies	miRNA sponges, anti-miRNA oligonucleotides, CRISPR/Cas9 emerging as precise EBV-targeted strategies.	Targeted therapies improving precision in EBV-related malignancies.	[126]

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
