# Peer review of "Immune Deficiency/Dysregulation-Associated EBV-Positive Classic Hodgkin Lymphoma"

_cancers, 2025, doi:10.3390/cancers17091433_

Round 1

Reviewer 1 Report

Comments and Suggestions for Authors

This review is interesting regarding the frequent presence of EBV in the diagnosis of Lymphoma; however, some points should be clarified:

  1. The abbreviated term CHL hides the main subject of this review, I suggest that the whole term review should be written.
  2. The authors suggest changes of classification which could not be included in a review (P14,p3.L1-22).
  3. The term “classic Hodgkin lymphoma” used several times along the text should be explained and differentiations with “non-classic” be also explained.
  4. The line “Refining diagnosis and tailoring management” looks strange in a review. Does this line represent selection bias?  The presence of EBV genes is found in some patients, is this proportion related with improved therapeutic results?

Author Response

1st reviewer comments:

Replies:

The abbreviated term CHL hides the main subject of this review, I suggest that the whole term review should be written.

Thank you for your comment. We have updated the title to spell out the full term

The authors suggest changes of classification which could not be included in a review (P14,p3.L1-22).

Thank you for your observation. Rewording of the criticized sentences has been carried out. Indeed, the statement that EBV-positive cases, particularly in immunosuppressed populations, may have worse prognosis due to immune evasion and treatment resistance is suggested by the WHO classification (see also Abstract, lines 13-14). It is not this review that suggests changes of classification.

The term “classic Hodgkin lymphoma” used several times along the text should be explained and differentiations with “non-classic” be also explained.

Thank you for your observation. We have now clarified the term “classic Hodgkin lymphoma” in the Introduction and provided a concise explanation distinguishing it from nodular lymphocyte-predominant Hodgkin lymphoma

The line “Refining diagnosis and tailoring management” looks strange in a review. Does this line represent selection bias?  The presence of EBV genes is found in some patients, is this proportion related with improved therapeutic results?

The line “Refining diagnosis and tailoring management” has been removed from the title.

Reviewer 2 Report

Comments and Suggestions for Authors

The manuscript entitled: “Immune deficiency/dysregulation-associated EBV positive cHL. Refining diagnosis and tailoring management (ID: cancers-3585006)” by Alibrahim et al. aims to explore the potential causal link between viral infections (e.g. EBV, HIV and hepatitis virus) and the development of cHL to understand their contribution in cHL pathogenesis.

Albeit the review is well written and prepared, comments should be addressed to further improve the manuscript.

Comments:

  1. Title: Abbreviations should (albeit commonly well known) be avoided within the title.
  2. Page 5. These both paragraphs are redundant and should be rephrased.
  3. Section: management: for a better overview please structure this section in e.g. First-line treatment, relapsed/refractory and non-transplant eligible, younger/advanced age.
  4. Table 3: Key outcomes: For a better overview and to make it more comparable, please provide detailed information with metrics within this section (e.g. “strong potential” should be more defined).
  5. Please check the whole manuscript for typing errors.
  6. Moreover, please provide references after each section where appropriate.

Author Response

2nd reviewer comments:

Replies:

Title: Abbreviations should (albeit commonly well known) be avoided within the title.

Thank you for the suggestion. We have revised the title.

Page 5. These both paragraphs are redundant and should be rephrased.

According to the Reviewer suggestion, the two paragraphs preceding Table 1 have been rephrased

Section: management: for a better overview please structure this section in e.g. First-line treatment, relapsed/refractory and non-transplant eligible, younger/advanced age.

Thank you for this helpful suggestion. We have restructured the "Management" section to provide a clearer overview

Table 3: Key outcomes: For a better overview and to make it more comparable, please provide detailed information with metrics within this section (e.g. “strong potential” should be more defined).

Thank you for your valuable feedback. We have revised Table 3 to include specific outcome metrics (e.g., CR rate, 2-year PFS/OS) where available

Please check the whole manuscript for typing errors.

Thank you for your observation. We have carefully reviewed the entire manuscript and corrected all identified typographical and formatting errors.

Moreover, please provide references after each section where appropriate.

Thank you for your valuable comment. We have now reviewed the manuscript thoroughly and added appropriate references

Reviewer 3 Report

Comments and Suggestions for Authors

There are some comments.

It would be better to consider analyzing the association between EBV and cHL by tumor site and geographical region, as these factors may influence EBV prevalence and clinical features.

It would be helpful to compare the histological features of EBV-positive versus EBV-negative cHL.

It would be helpful to consider adding EBV gene expression patterns across EBV-associated hematolymphoid neoplasms (e.g., latency types I–III).

The conclusion would benefit from being more concise and focused on the main findings and clinical implications.

Including a list of abbreviations is recommended for clarity and ease of reference.

Comments on the Quality of English Language

Please check consistency of terminology, English grammar, and spelling
 For example,  Immune deficiency/dysregulation associated EBV positive cHL.
                      -> Immune deficiency/dysregulation-associated EBV positive cHL.

Author Response

3rd reviewer comments:

Replies:

It would be better to consider analyzing the association between EBV and cHL by tumor site and geographical region, as these factors may influence EBV prevalence and clinical features.

We thank the reviewer for the suggestion. This point has been addressed in the revised Introduction

It would be helpful to compare the histological features of EBV-positive versus EBV-negative cHL.

We appreciate the suggestion. A comparison of histological features between EBV-positive and EBV-negative cHL has been added to the revised Differences Between EBV-positive and EBV-negative cHL section.

It would be helpful to consider adding EBV gene expression patterns across EBV-associated hematolymphoid neoplasms (e.g., latency types I–III).

Thank for your suggestion. In the revised Table 2 latency types have been added

The conclusion would benefit from being more concise and focused on the main findings and clinical implications.

Thank you for your valuable suggestion. In response, we have revised the conclusion to make it more concise and directly focused on the key findings and clinical implications

Including a list of abbreviations is recommended for clarity and ease of reference.

Thank you for the helpful suggestion. We have now included a comprehensive list of abbreviations used throughout the manuscript to enhance clarity and facilitate ease of reference for readers.

Please check consistency of terminology, English grammar, and spelling
 For example,  Immune deficiency/dysregulation associated EBV positive cHL.
                      -> Immune deficiency/dysregulation-associated EBV positive cHL.

We appreciate your feedback regarding the consistency of terminology, grammar, and spelling. The manuscript has been thoroughly reviewed and revised to ensure accurate and consistent use of hyphenation and terminology, including corrections such as “immune deficiency/dysregulation-associated EBV-positive cHL.”